# atomSmltr: a modular Python package to simulate laser cooling setups.

**Mateo Weill[1,2], Andrea Bertoldi [2] and Alexandre Dareau [1,2⋆]**

**1** Institut d'Optique, F-33400 Talence, France
**2** IOGS, LP2N, Université Bordeaux, CNRS, UMR 5298, F-33400 Talence, France

⋆ alexandre.dareau@institutoptique.fr

## Abstract

We introduce atomSmltr, a Python package for simulating laser cooling in complex magnetic field and laser beams geometries. The package features a modular design that enables users to easily construct experimental setups, including magnetic fields, laser beams and other environment components, and to perform a range of simulations within these configurations. We present the overall architecture of atomSmltr and illustrate its capabilities through a series of examples, including benchmarks against standard textbook cases in laser cooling.

## 1 Introduction

Atom-light interactions lie at the core of many fundamental phenomena and underpin a wide range of applications [1]. The fact that the absorption or emission of a photon affects an atom's motion sparked a new direction in experimental atomic physics, ultimately giving rise to the field of cold atoms [2, 3]. Experiments with cold atoms have profoundly influenced atomic physics, enabling advances in quantum simulation [4], precision metrology [5], and quantum information [6].

While the effects of a resonant laser on atomic motion can be analytically described in simple cases, more complex configurations, involving multiple laser beams with several parameters (polarization, direction, detuning, etc.) or complex magnetic field landscapes, require numerical simulations. A collection of atomic physics oriented software packages are available in various programming languages [7–11]; nevertheless, a user-friendly Python library

specifically focused on laser cooling remains lacking[1]. This gap motivated the development of `atomsmltr`, a modular and accessible Python library dedicated to simulating laser cooling. In this paper, we introduce `atomsmltr`, present its architecture, and provide a series of examples to demonstrate its capabilities and benchmark its performance on canonical laser cooling scenarios. The paper is organized as follows:

- in section 2, we present the general approach of `atomsmltr`, including its intended purpose and the main underlying physical assumptions;

- in section 3, we describe the specific implementation of `atomsmltr`, detailing the global architecture of the package and the role of its main components;

- in section 4, we benchmark the package using canonical laser cooling scenarios;

- in section 5 we demonstrate how `atomsmltr` can be applied to more complex situations, providing insights into the physics of laser cooling.

The appendices provide additional information to help users get started with `atomsmltr` (appendix A), offer a brief comparison with other available software (appendix B), and clarify conventions for laser propagation and polarization (appendix C).

---

☞ **Disclaimer: How to read this paper**

This paper is intentionally written in a comprehensive way, providing detailed information on the code architecture, code examples, and illustrations of physically relevant configurations. It is intended both as a quick user guide for the `atomsmltr` package, and as a demonstration of the module's performance and capabilities through canonical laser cooling examples. Most sections can be read independently, depending on the reader's interest.

---

➥ **Try it!**

The most effective way to understand `atomsmltr` is to use it. The package can be installed via pip[2]:

```
pip install atomsmltr
```

For further information and documentation, please visit:

https://atomsmltr.readthedocs.io.

---

[1]See appendix B for a non-exhaustive list of available packages, and a quick comparison with `atomsmltr`.

[2]All examples provided in this paper use version `0.1.5` of `atomsmltr`, which was the latest release available on PyPi at the time of writing.

## 2 Package description

### 2.1 Main approach: Python and modular

The goal of `atomsmltr` is to provide a straightforward numerical tool for simulating laser cooling in complex laser and magnetic field configurations, for example to optimize a cold-atom source. We chose Python because it is widely used in the physics community, which facilitates adaptation of the code to specific user needs. As detailed below, the package is built with a modular architecture that separates the different stages involved in simulating and optimizing a complex cold-atom system: defining lasers and magnetic field parameters, configuring the overall experimental setup, and performing targeted simulations on the system.

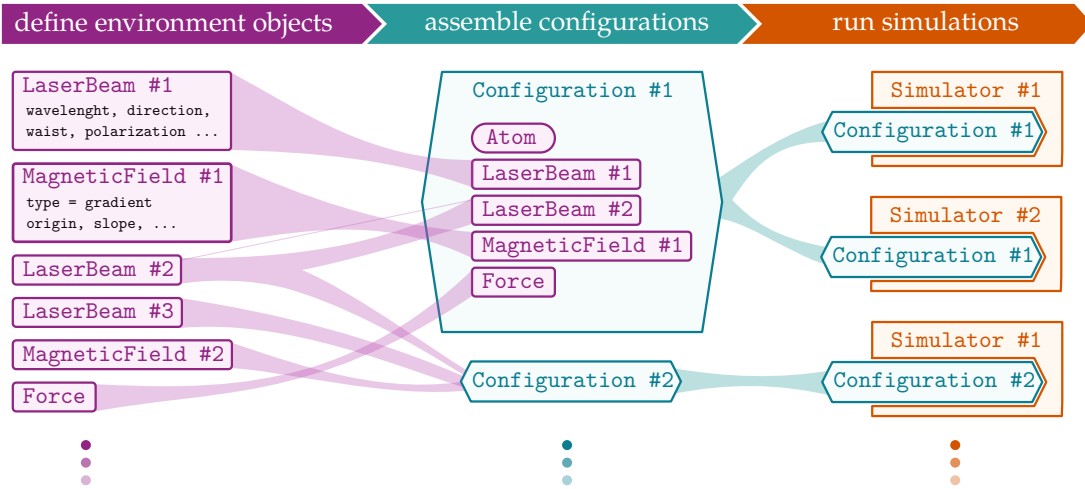

Figure 1: Typical simulation workflow with `atomsmltr`. Users begin by defining a set of environment objects (*e.g.*, laser beams, magnetic fields, forces), and then combine them to create experimental configurations. These configurations are then passed to simulators to perform the simulations. The modular architecture makes the package user-friendly and allows seamless extension with new features, such as additional types of environment objects or simulators.

The overall architecture of `atomsmltr` is illustrated in Figure 1. A typical simulation proceeds in three steps. First, the user defines a set of environment objects (*e.g.*, laser beams, magnetic fields, forces) that interact with the atoms. Second, these objects are encapsulated within a configuration object. Third, the configuration is passed to a simulator. The modular design of `atomsmltr` ensures that each step is largely independent, which simplifies the recombination of environment objects into multiple configurations and enables the execution of diverse simulations for each configuration.

We anticipate that this modular design will allow users to extend `atomsmltr` by integrating new tools tailored to their specific needs (*e.g.*, additional simulation types, magnetic field configurations, or laser beam profiles) while still leveraging the package's existing features. In this context, we have also aimed to integrate `atomsmltr` into a broader ecosystem of physics-oriented simulation tools. For example, it is straightforward to include magnetic fields generated with the `magpylib` package [12] within `atomsmltr` configurations.

## 2.2   Simulation target

The `atomsmltr` package is designed to simulate the interaction of a single neutral atom with a set of magnetic fields and laser beams in a three-dimensional environment. In this context, `atomsmltr` can be efficiently used to:

- simulate the effect of radiation forces on an atom within complex laser beam and magnetic field configurations;

- propagate a collection of atoms with different initial conditions in a given experimental setup, leveraging optimized array operations;

- define complex experimental configurations including lasers, magnetic fields (also created with `magpylib`), forces and zones, for subsequent simulations, as well as for computing local forces, intensities, magnetic fields and polarizations;

- handle the local decomposition of a laser polarization on atomic transitions ($\sigma^{\pm}$, $\pi$) for arbitrary laser directions and polarizations in complex magnetic field landscapes.

Users should be aware that `atomsmltr` currently has the following limitations:

- It does not include internal atomic structure effects beyond Zeeman splitting and the selection rules for $J = 0 \rightarrow 1$ transitions; phenomena such as optical pumping and electromagnetically induced transparency (EIT), are therefore not captured by the current model.

- It does not account for off-resonant lightshifts (*e.g.*, those arising from dipole traps).

- Magnetic trapping of atoms is not included.

- Interference effects between laser beams are neglected.

- Atom-atom interactions and collective effects are not considered.

Most of these limitations arise from our deliberate choice to maintain a simplified framework for atom-light interactions, which is sufficient for simulating, designing, and optimizing systems such as magneto-optical traps or cold-atom sources. This choice was primarily motivated by performance considerations, as including more refined models such as optical Bloch equations would significantly increase the computational overhead. However, the modularity of the package allows for the future incorporation of new `Simulation`[3] objects, enabling additional use cases. This approach builds on the versatility of the existing `Configuration` and environment classes, supporting a broader range of simulations.

## 2.3   Physical hypotheses

### 2.3.1   Single-particle physics

`atomsmltr` is currently limited to single-particle physics; therefore, effects such as interatomic collisions, photon screening, and multiple scattering are not included in the simulations.

---

[3]See section 3 for details on the `atomsmltr` classes and objects.

### 2.3.2 Atom-light interactions

In its current stage of development, `atomsmltr` models the atom in terms of $J = 0 \rightarrow 1$ transitions, representing the minimal framework necessary to simulate phenomena such as Zeeman slowing or trapping in a magneto-optical trap. An atom illuminated by a near-resonant laser beam experiences a mean force known as radiation pressure, given by [13]:

$$\vec{F}_{\text{rad}} = \hbar \vec{k} \frac{\Gamma}{2} \frac{s}{1+s}, \tag{1}$$

where $\vec{k}$ is the laser wavevector, $\hbar$ the reduced Planck constant and $\Gamma$ the natural linewidth of the atomic transition. The saturation parameter $s$ is defined as:

$$s = \frac{I}{I_{\text{sat}}} \frac{1}{1 + (2\delta/\Gamma)^2}, \tag{2}$$

where $I$ is the laser intensity at the atom's position, $I_{\text{sat}}$ the saturation intensity associated with the transition, and $\delta$ the laser detuning from resonance. For a set of laser beams, we assume that the total force exerted on the atom can be approximated as the sum of the individual forces from each beam, which is valid in the weak-saturation regime ($s \ll 1$). More refined approaches, such as the so-called "rate-equation model" used in [9], are not yet implemented in `atomsmltr`.

### 2.3.3 Stochastic simulations

In some cases, it is useful to account for stochastic effects arising from the Poissonian statistics of photon absorption and spontaneous emission. To this end, specialized `Simulation` objects are provided to implement these stochastic effects. The current implementations assume that a sufficiently large number of photons are scattered during each integration step, which justifies the use of a random-walk model with Gaussian fluctuations, without explicitly drawing and summing the contribution of each scattered photon.

More specifically, an atom that absorbs on average $\langle N \rangle = N_0$ photons during an integration step of duration $\Delta t$ experiences a mean radiative force $\langle \vec{F}_{\text{rad}} \rangle = \hbar \vec{k} N_0 / \Delta t$. Fluctuations around this mean value are included through an additional random force $\delta \vec{F}_{\text{rad}} = \hbar \vec{k} \delta N / \Delta t$, where, for large $N_0$, $\delta N$ can be approximated as a Gaussian variable with zero mean and standard deviation $\sigma = \sqrt{N_0}$. Spontaneous emission is modeled by another random force $\delta \vec{F}_{\text{em}} = (\hbar k / \Delta t) \delta \vec{u}$, where $\delta \vec{u} = (\delta N_x, \delta N_y, \delta N_z)$ is a random vector whose components $\delta N_{i=x,y,z}$ are Gaussian variables with zero mean and standard deviation $\sigma = \sqrt{N_0/3}$.

## 3 Implementation

In this section, we provide more details on the architecture of `atomsmltr`. Readers interested in practical code examples demonstrating the use of `atomsmltr` may refer to the code snippet provided in the appendix, section A.3.

### 3.1 Package overview

The `atomsmltr` package provides several classes that enable users to select an atomic species, define an experimental configuration composed of various of environmental objects (*e.g.*, laser beams, magnetic fields, forces, zones), and perform simulations within this setup. The main submodules are summarized as follows:

- `atomsmtlr.atoms` : defines the generic `Atom` class, and includes a collection of ready-to-use atomic species (`Ytterbium`, `Strontium`, `Rubidium`). It also implements the `Transition` class, which encapsulates atom-light interaction formulas, such as those used to compute scattering rates.

- `atomsmtlr.environment` : contains classes for describing the experimental setup, including `LaserBeam`, `Force`, `MagneticField` and `Zone`. These are *generic* base classes, each with specific implementations, like, for example, Gaussian laser beams, plane waves, magnetic field gradients or quadrupole configurations.

- `atomsmtlr.simulation` : provides the `Configuration` class to combine environment objects and define atom-light interaction parameters for a given experimental setup, and the `Simulation` class to perform physical simulations within this configuration. Several types of simulations are implemented, see section 3.5 for details.

- `atomsmtlr.examples` : offers a set of predefined configurations for quickly testing or benchmarking `atomsmltr`. All examples presented in this work are available in this submodule (see also appendix A.4).

## 3.2 Atoms

```
from atomsmltr.atoms import Ytterbium, Strontium, Rubidium
```

The `atom` submodule implements all the classes required to define atomic species and compute their properties. The `Atom` class allows users to create an atomic species by specifying its physical parameters and associated list of atomic transitions. These transitions are managed by the `Transition` class, which provides methods to compute relevant physical quantities such as the scattering rate, the Doppler temperature, and saturation parameter. Currently, as discussed in the previous section, only $J = 0 \rightarrow 1$ transitions are implemented.

The `atom` submodule also provides a small collection of predefined atomic species (ytterbium, strontium and rubidium), each including a limited subset of transitions. This collection will be expanded in future releases. Users can also define new atomic species or transitions at runtime by instantiating an empty `Atom` or `Transition` object and manually specifying the relevant properties.

## 3.3 Environment objects

```
from atomsmltr.environment.lasers import GaussianLaserBeam
from atomsmltr.environment.lasers.polarization import CircularLeft
from atomsmltr.environment.fields import MagneticOffset
from atomsmltr.environment.zones import Box
```

The `environment` submodule implements a range of classes defining the elements that shape the atom's environment, such as laser beams or magnetic fields. All environment objects share a common structure, which facilitates their combination when constructing simulation configurations. For example, each environment object includes a `.tag` property, either user-defined or automatically generated, that serves as a unique identifier when multiple objects are assembled within a configuration. Environment objects also return position-dependent values, either scalar of vectorial, through a standardized `.get_value()` method. In addition, they provide a convenient `.print_info()` method for quick inspection of key properties, as well as a collection of plotting utilities (see the documentation for further details).

### 3.3.1 Laser beams

```python
from atomsmltr.environment.lasers import GaussianLaserBeam
from atomsmltr.environment.lasers.polarization import CircularLeft

beam = GaussianLaserBeam(
    wavelength=399e-9,
    waist=50e-6,
    power=30e-3,
    waist_position=(0, 0, 0),
    direction=(0, 0, 1),
    polarization=CircularLeft(),
)
```

Laser beams are defined within the `environment.lasers` submodule, which provides a set of classes for specifying laser beams with customizable parameters such as waist position and size, propagation direction, and intensity profile. A dedicated class is included to handle laser polarization, enabling the computation of polarization decomposition onto atomic transitions in complex laser and magnetic field configurations. Details of the polarization conventions and the model used to describe laser properties are provided in Appendix C. Currently, two types of laser beams are implemented: `GaussianLaserBeam`, representing an ideal Gaussian beam, and `PlaneWaveLaserBeam`, describing a plane wave with uniform intensity and infinite transverse extent.

### 3.3.2 Fields: magnetic fields and forces

```python
import numpy as np
from atomsmltr.environment import ConstantForce
from atomsmltr.atoms import Ytterbium
from atomsmltr.environment.fields import MagneticOffset

m = Ytterbium().mass  # kg
g = 9.81  # m/s^2
direction = np.array([0, 0, -1])  # along -z
grav_force = m * g * direction

gravity = ConstantForce(field_value=grav_force, tag="gravity")
mag_offset = MagneticOffset(field_value=(0,1,0), tag="offset")
```

The `environment.fields` submodule provides a framework for implementing vector fields. It currently includes two submodules: `..fields.magnetic` and `..fields.force`, which handle magnetic fields and external forces, respectively. Several idealized magnetic field profiles are available, including `MagneticGradient`, `MagneticOffset` and `MagneticQuadrupole`. More realistic profiles can also be adopted by supplying numerical one-dimensional or three-dimensional data to the `InterpMag1D1D` and `InterpMag3D3D` classes, which interpolate the input field maps (see the documentation for details).

A notable feature of `atomsmltr` is its seamless integration with `magpylib` [12], enabling the use of realistic magnetic fields generated by configurations of electromagnetic coils and permanent magnets, as illustrated below. However, computing complex magnetic field geometries with `magpylib` can introduce significant computation overhead. For this reason,

we recommend using the built-in idealized magnetic field profiles whenever they sufficiently capture the relevant physics.

```python
import magpylib as magpy
from atomsmltr.environment.fields.magnetic.magpylib import
    MagpylibWrapper

# create a magpylib object
cyl = magpy.magnet.Cylinder(
    polarization=(0.5, 0.5, 0),
    dimension=(40, 20),
)

# wrap it up
mag_field = MagpylibWrapper(cyl, tag="a nice magnet")
mag_field.plot2D(
    plane="XY",
    limits=(-50, 50, -50, 50),
    Npoints=(100, 101),
)
```

### 3.3.3 Zones

The `environment.zones` submodule defines spatial zones, either in position or velocity space, that can be used to terminate a simulation when an atom enters a specified region. Zones can also be employed to label atoms at the end of a simulation, for example to determine whether an atom has been captured in a magneto-optical trap. `Zone` objects can be combined using standard mathematical operators, allowing users to construct complex logical conditions. Currently implemented zone types include one-dimensional boundaries (`UpperLimit`, `LowerLimit`, `Limits`) and three-dimensional volumes (`Box`, `Cylinder`). Additional details and usage examples are provided in the package documentation.

## 3.4 Configuration

```python
from atomsmltr.simulation import Configuration
```

Once defined, environment objects can be assembled into a `Configuration` object, which can then be passed to a simulator. The `Configuration` class serves two primary purposes:

- it manages collections of environment objects, allowing users to easily construct and modify experimental setups;

- it defines atom-light interactions by specifying which lasers are coupled to which atomic transitions, along with the associated interaction parameters (*e.g.*, the detuning).

For the first task, the `Configuration` class provides a convenient set of methods to add, list, remove or update environment objects. Once an object is added, the configuration stores a copy of it; later modifications to the original environment object are therefore not automatically propagated. To apply such changes, the corresponding object must be explicitly updated within the configuration. All operations rely on the environment object's `.tag` property, which serves as its unique identifier. The `Configuration` class also includes a

`.print_info()` method that produces a human-readable summary of the configuration. Environment objects are internally organized by class type, allowing multiple categories of objects to be included simultaneously. An example illustrating how to add objects to a configuration using a simple mathematical operator is shown below.

```python
from atomsmltr.simulation import Configuration
from atomsmltr.atoms import Ytterbium
from atomsmltr.environment import MagneticOffset, GaussianLaserBeam

# define objects
laser1 = GaussianLaserBeam(tag="laser1")
laser2 = GaussianLaserBeam(tag="laser2")
mag_offset = MagneticOffset((0,1,0), tag="offset")

# init a configuration for ytterbium atoms
config = Configuration(atom=Ytterbium())

# add objects
config += laser1, laser2, mag_offset
```

The second role of the `Configuration` object is to define the interactions between lasers and atomic transitions, using the `add_atomlight_coupling()` method. This design choice reflects the fact that the wavelength specified in `LaserBeam` object is intended only to determine the spatial propagation of the beam. More detailed properties, such as the precise detuning relative to a given atomic transition, are defined within the `Configuration` object, and must be specified separately for each laser beam[4]. It is the user's responsibility to ensure that all laser-transition couplings are properly defined before running a simulation. When setting up these coupling, lasers and transitions are identified via their respective `.tag` attributes. A minimal example illustrating the creation of a configuration containing an atom and a laser, and the definition of their atom-light interaction, is shown below.

```python
from atomsmltr.simulation import Configuration
from atomsmltr.atoms import Ytterbium
from atomsmltr.environment import PlaneWaveLaserBeam

# init a configuration for ytterbium atoms
config = Configuration(atom=Ytterbium())

# get ytterbium main transition information
main = config.atom.trans["main"]

# create a plane wave laser
laser = PlaneWaveLaserBeam()
laser.wavelength = main.wavelength
laser.set_power_from_I(main.Isat)   # set power to have I=Isat
laser.tag = "399"

# add laser to config
config += laser
```

```
# create a coupling
config.add_atomlight_coupling(
    laser="399",
    transition="main",
    detuning=-2 * main.Gamma,
)

# show info
config.print_atomlight_info()
```

## 3.5  Simulations

```
from atomsmltr.simulation import ScipyIVP_3D, RK4, Euler, EulerSt
```

The final step in performing a simulation with `atomsmltr` is to pass the `Configuration` object to a `Simulation` class. Instead of offering a single simulation class with numerous parameters to select the integration method or physical effects, `atomsmltr` adopts a modular "one class per method" design. This approach simplifies experimentation with different simulation techniques and enables users to implement custom simulation classes, albeit at the cost of a larger number of available classes. All `Simulation` classes share a common interface. The most important method is `.integrate()`, which performs the simulation for a specified set of initial conditions and time steps. Additionally, `Simulation` provides a `.run()` method, which distributes a list of initial conditions across multiple threads to enable parallel computation (see documentation for details).

In the current version of `atomsmltr`, the following simulators are implemented:

- **"Deterministic" integrators**, which do not account for random fluctuations arising from photon absorption and spontaneous emission (cf. section 2.3.3):

    - `ScipyIVP_3D`: a wrapper for the `scipy` solver `solve_ipv` [5]. This solver supports several integration methods (see the documentation) and is generally more efficient than the other built-in solvers, as it adapts the integration steps automatically. However, unlike the other simulation classes in `atomsmltr`, `ScipyIVP_3D` does not support vectorization of the initial conditions (see section 3.7).
    - `Euler`: a simple Euler method solver.
    - `RK4`: a solver based on the fourth order Runge-Kutta method.
    - `VelocityVerlet`: a solver implementing the velocity Verlet method.

- **"Stochastic" integrators**, which include random fluctuations due to photon absorption and spontaneous emission: `EulerSt` and `RK4St`, corresponding to Euler and fourth order Runge-Kutta methods, respectively.

Examples of how to perform simulations are provided below (see, for instance, section A.3) and in the online `atomsmltr` documentation.

---

[4]In a future version of `atomsmltr`, far-off-resonant lightshifts could be implemented in a similar fashion through an `add_offresonant_lightshift()` method. This approach would allow the simulation to distinguish which lasers beams are used to compute near-resonant scattering and which contribute to off-resonant lightshifts. Such an extension would broaden the capabilities of `atomsmltr`, enabling its application to dipole-trap physics.

[5]https://docs.scipy.org/doc/scipy/reference/generated/scipy.integrate.solve_ivp.html

## 3.6  Examples

```
from atomsmltr.examples.atomic_fountain import config, description
print(description)
```

The `atomsmltr` package includes a collection of ready-to-use configuration examples, designed to facilitate testing and experimentation. Those are provided in the `examples` submodule. Each example module contains one or more built-in configuration objects, along with a `description` string. In the current version, the provided examples correspond to the scenarios discussed in this paper:

- `examples.chen2021`: examples taken from the `AtomECS` paper [9], used here to benchmark `atomsmltr` results. This module includes:

   > `config_1D_molasses`, corresponding to a one-dimensional optical molasses with rubidium, shown in Figure 4(a),

   > `config_1D_MOT`, corresponding to a one-dimensional magneto-optical trap for strontium as in Figure 4(b).

- `examples.atomic_fountain`: configuration for a rubidium atomic fountain in the (1,1,1) arrangement, used as an application example in section 5 (Figure 7).

- `examples.feng2024`: collection of several configurations for a strontium source as described in [14], also used as an application example in section 5 (Figure 6). The configuration used for the example in Figure 6 is `config_asymmetric_field_2`

- `examples.atomsmtlr2025`: contains other examples used in the present paper to benchmark `atomsmltr`. This module includes:

   > `config_1D_MOT_Yb`, a one-dimensional ytterbium MOT configuration used for the physics benchmark of Figure 2,

   > `config_3D_MOT_Yb`, a three-dimensional ytterbium MOT configuration used for the performance benchmark of Figure 5,

   > `config_Doppler_limit`, a three-dimensional optical molasses configuration for ytterbium atoms used for the physics benchmark of Figure 3.

## 3.7  Spatial coordinates convention and vectorization

In `atomsmltr` high-level methods, positions are expressed in Cartesian coordinates in the laboratory frame. This is, for example, the case for the `get_value()` shared by all environment objects. To leverage NumPy's array parallelization, methods that take spatial coordinates as input follow a common vectorization convention, inspired by the magpylib package [12]. In this convention, spatial coordinates are passed as a NumPy array of shape (,3) or (n, m, ..., 3), where the last axis always has dimension three and contains the Cartesian coordinates $(x, y, z)$ in the laboratory frame. Correspondingly, the returned values have shape (,i) or (n, m, ..., i), where the last axis contains the function output (i=1 for a scalar function, i=3 for a vector function).

The same convention applies to the initial conditions of the simulation, which are represented as arrays of shape (,6) or (n, m, ..., 6), with the last axis containing Cartesian position and velocities in the laboratory frame: $(x, y, z, v_x, v_y, v_z)$. Using this convention, all operations involved in the integration process, from evaluating environment object values to propagating differential equations, can be fully vectorized. Consequently, the evolution of a collection of $N$ atoms with different initial conditions in a given configuration can be computed efficiently (see section 4.2 for details).

# 4 Benchmark

## 4.1 Physical benchmark

In the following section, we benchmark simulations performed with `atomsmltr` across a set of atom-cooling scenarios. We compare the results with analytical solutions as well as with outputs obtained from other established simulation tools.

### 4.1.1 Notations

Unless otherwise specified, the following notation will be used for the physical quantities appearing in the examples:

- $\hbar$: the reduced Planck constant,
- $k_B$: Boltzmann's constant,
- $m$: the atom's mass (kg),
- $\lambda$: the considered transition wavelength (m),
- $k = 2\pi/\lambda$: the considered transition wavenumber (m$^{-1}$),
- $I_{\text{sat}}$: the considered transition saturation intensity (W m$^{-2}$),
- $\Gamma$: the considered transition natural linewidth (s$^{-1}$),
- $\delta$: the detuning of the laser with respect to the atomic transition, negative for a red-detuned laser (rad s$^{-1}$),
- $I_0$: the laser intensity, at focus for a Gaussian beam (W m$^{-2}$),
- $s_0 = I/I_{\text{sat}}$: the laser saturation parameter, at focus for a Gaussian beam,
- $b$: in magneto-optical traps, the magnetic field gradient along the weak axes (T m$^{-1}$).

### 4.1.2 One-dimensional MOT: damped harmonic oscillator

We begin by examining the motion of an atom in a one-dimensional magneto-optical trap (1D MOT), which provides a minimal yet sufficiently representative system for benchmarking our code. The setup consists of two counter-propagating, circularly polarized laser beams that are red-detuned with respect to the atomic transition, in combination with a magnetic field gradient. We denote the atom's position and velocity by $z$ and $v_z$, respectively. In the regime where the Doppler shift is negligible with respect to the natural linewidth of the transition, *i.e.*, $kv_z \ll \Gamma$, the radiative forces acting on the atom can be linearized [15]. Under these conditions, the total force along the $z$-axis in the 1D MOT configuration can be written as:

$$F_{z,\text{rad}} = -m\gamma v_z - \kappa z, \tag{3}$$

where $\gamma$ and $\kappa$ are, respectively, the friction coefficient and the harmonic restoring constant, given by [15]:

$$\gamma = -\frac{\hbar k^2}{m} s_0 \frac{2\delta\Gamma}{\delta^2 + \Gamma^2/4}, \tag{4}$$

$$\kappa = -k\mu b s_0 \frac{2\delta\Gamma}{\delta^2 + \Gamma^2/4}, \tag{5}$$

where $\mu$ is the magnetic moment of the atom in the excited state (see section 4.1.1 for the definition of the other parameters).

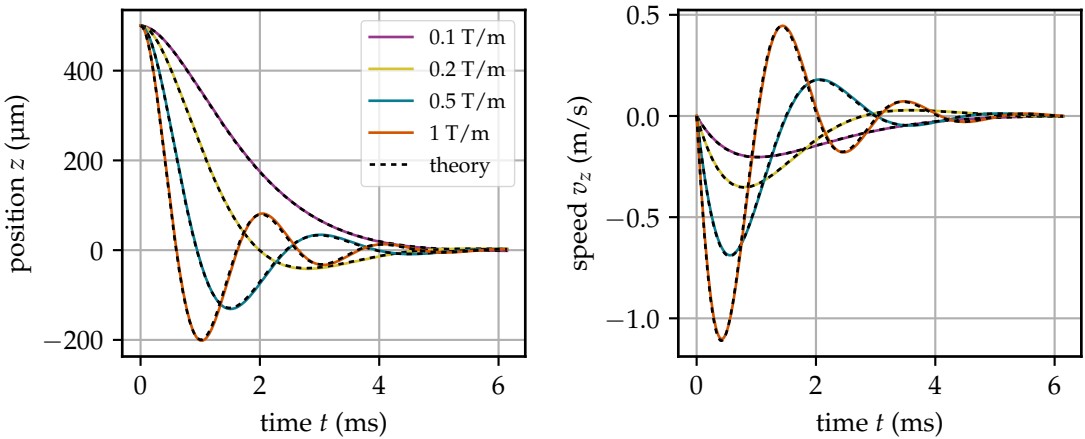

Figure 2: Motion of an ytterbium atom in a 1D MOT, simulated using `atomsmltr` (solid lines) and compared to analytical predictions (black dotted lines). Results are shown for several magnetic field gradients $b$ (see legend). The atom exhibits damped oscillatory motion, with the oscillation frequency increasing with $b$, in good agreement with theoretical expectations.

Using Eqs. (4) and (5), we simulate the motion of an atom and compare the results to those obtained with `atomsmltr`. Figure 2 shows the tajectories of an ytterbium atom in a 1D MOT operating on the main transition ($\lambda = 399$ nm). The simulations are performed with a saturation parameter $s_0 = 0.02$ and a detuning $\delta = -\Gamma/2$. The atom is initially placed at rest 500 µm from the magnetic field zero, and the simulation is repeated for several values of the magnetic field gradient $b$. For simplicity, plane wave laser beams are assumed. Under these conditions, the analytical prediction and numerical results show excellent agreement.

### 4.1.3 Stochastic evolution: Doppler cooling limit

Another textbook example of laser cooling is the three-dimensional optical molasses configuration, consisting of three pairs of counter-propagating, red-detuned laser beams, aligned along the Cartesian axes and operated in a homogeneous, zero magnetic field. By accounting both for the radiative pressure forces and for the fluctuations arising from the Poissonian nature of photon scattering, one can analytically derive the steady-state temperature of the atom, known as the Doppler temperature $T_{\text{Doppler}}$, which is given by [15]:

$$T_{\text{Doppler}}(\delta) = \frac{\hbar}{2k_B} \frac{\delta^2 + \Gamma^2/4}{|\delta|}. \tag{6}$$

The final temperature depends on the laser detuning $\delta$, and reaches a minimum value of $T_0 = \hbar\Gamma/2k_B$ for $\delta = -\Gamma/2$. When cooling on a $J = 0 \rightarrow 1$ atomic transition, where sub-Doppler mechanisms are absent, the Doppler temperature represents a hard limit. It therefore provides a stringent benchmark for validating the stochastic simulation models implemented in `atomsmltr`.

In Figure 3 we present the results of a simulation of Doppler cooling in a three-dimensional optical molasses for ytterbium atoms. Cooling is performed on the main transition ($\lambda = 399$ nm), corresponding to a minimal Doppler temperature of $T_0 = 693$ µK. Since optical molasses provides no confinement, the atoms undergo a random walk. To eliminate artifacts arising from the finite size of Gaussian laser beams, the simulation uses plane waves of infinite extent. Each beam has an intensity of $I_0 = 0.05\,I_{\text{sat}}$, and the detuning

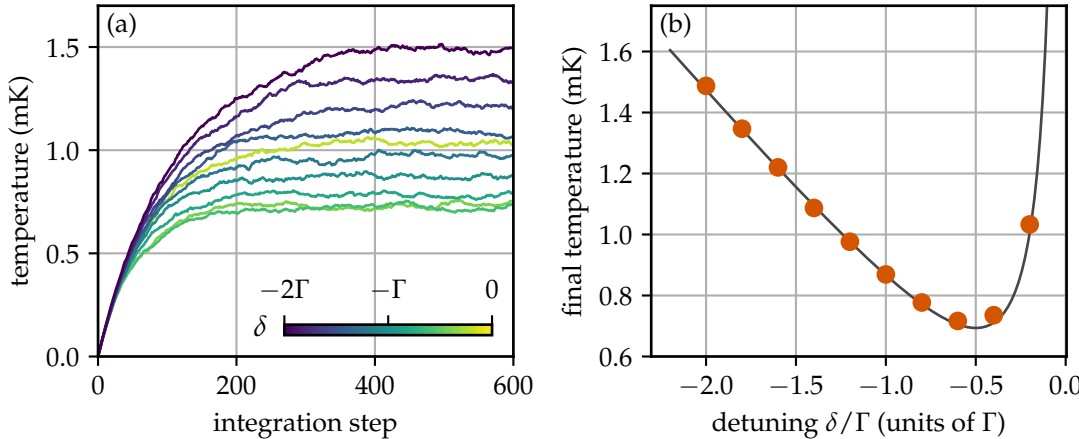

Figure 3: Simulation of the Doppler cooling limit in three-dimensional optical molasses for ytterbium atoms on the main transition ($\lambda = 399\,\text{nm}$). An ensemble of $2 \times 10^3$ atoms is initialized at rest and allowed to thermalize for various lasers detuning $\delta$. **(a)** Time evolution of the ensemble temperature. The total integration time is adjusted for each $\delta$ to maintain an approximately constant number of scattered photons. The temperature reaches a steady-state plateau by the end of the simulation. **(b)** Final temperature as a function of the laser detuning $\delta$, averaged over the last 150 simulation steps, corresponding to the steady-state regime. Simulation results (circles) are compared with the analytical prediction (solid line) given by Equation (6).

$\delta$ is varied between $-0.2\,\Gamma$ and $-2\,\Gamma$. For each $\delta$, simulations are performed with an ensemble of $2 \times 10^3$ atoms, leveraging vectorization (see section 3.7) and the `RK4St` integrator. The atoms are initially at rest at the origin of the laboratory frame. Each integration comprises 600 steps, with the total duration adjusted according to the detuning so that approximately $3 \times 10^3$ photons are scattered in total. To confirm the attainment of thermal equilibrium, the ensemble temperature is computed after each integration step (see Figure 3(a)), showing that a steady-state plateau is reached by the end of the simulation. A comparison between the simulated and analytical results (Figure 3(b)) shows excellent agreement, thereby validating our stochastic simulation model.

### 4.1.4 Comparison with atomECS: one-dimensional molasses and magneto-optical trap

We now turn to benchmarking `atomsmltr` in more complex scenarios, where analytical models are not readily available. To this end, we compare our results with those obtained using the `atomECS` module [9], a laser-cooling simulation tool written in Rust[6]. The comparison focuses on two configurations presented in [9] to demonstrate the capabilities of `atomECS`: (i) cooling dynamics in a one-dimensional optical molasses, and (ii) atom capture in a one-dimensional MOT, respectively with rubidium and strontium atoms.

The simulation results are presented in Figure 4, showing good agreement between `atomsmltr` and `atomECS`. For the optical molasses case (Figure 4(a)), the two simulations produce nearly identical results, with no visible discrepancy. In the case of atom capture in a one-dimensional MOT (Figure 4(b)), agreement is excellent both below and above the capture velocity limit. Near the capture limit, however, a mismatch appears between the two

---

[6]See appendix B for a brief comparison of `atomsmltr` with other existing solutions.

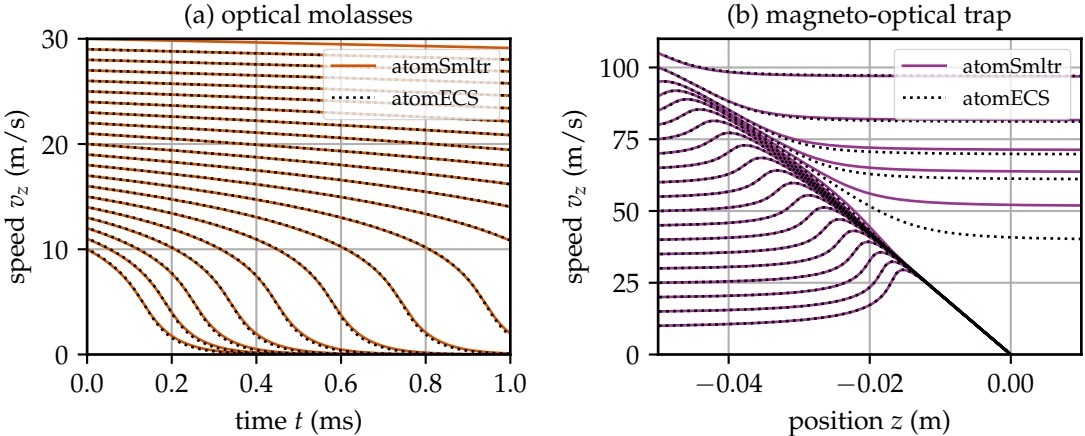

Figure 4: Comparison between `atomsmltr` (Python) and `atomECS` (Rust) using examples from [9]. **(a) Cooling dynamics in a one-dimensional optical molasses** for rubidium atoms on the D2 line at 780 nm (see main text for details). Excellent agreement is observed between the `atomsmltr` results (plain orange line) and the `atomECS` ones (dotted black line). **(b) Cooling and capture in a one-dimensional MOT** for strontium atoms on the main transition at 461 nm. Away from the capture limit, *i.e.*, for $v_z$ way above or bellow $80\,\mathrm{m\,s^{-1}}$, good agreement is found between `atomsmltr` (plain purple line) and `atomECS` (dotted black line). A visible discrepancy appears in the tail of the trajectory of nearly-captured atoms, which is highly sensitive to the exact point at which the atom leaves the cooling trajectory. This effect can depend on the integration method and time step, but has minimal impact on relevant quantities, such as the capture velocity limit.

simulations. This behavior, which persists across different integration methods, arises because atoms initially slowed by the MOT may deviate from the cooling trajectory and eventually be lost. Minor differences in the exact moment at which an atom is lost can lead to noticeable variations in its final velocity, explaining the observed mismatch. We emphasize that, although the trajectories of lost atoms differ, the inferred capture velocity is only minimally affected, which is sufficient for applications such as cold-atom setup optimization. In these examples, realistic Gaussian laser beams were used, and details of the simulation parameters are provided in appendix A.4. As noted in section 3.6, it is also possible to directly import the configurations used for these simulations from the `examples.chen2021` submodule.

## 4.2 Performance

We benchmark the simulation performance by comparing the built-in integrators, namely `Euler`, `EulerSt`, `RK4`, `RK4St` and `VelocityVerlet`, against our wrapper for `scipy`'s solver `solve_ipv` (`ScipyIVP_3D`). The benchmark is conducted using a three-dimensional magneto-optical trap configuration for an ytterbium atom. The setup consists of three orthogonal pairs of counter-propagating Gaussian laser beams at 399 nm, each with a waist of 22 mm and a total combined power of 100 mW. A perfect quadrupole magnetic field with a gradient of $30\,\mathrm{G\,cm^{-1}}$ along its weak axis is assumed. Simulations are performed over 15 ms with 1000 integration steps, with ensembles of atoms whose initial positions and velocities are randomly drawn within $\pm5$ mm and $\pm5\,\mathrm{m\,s^{-1}}$, respectively.

For each integration method, we perform simulations[7] with an increasing number of atoms,

---

[7]Simulations were performed on a HP EliteBook 840 laptop, with an Intel® Core™ Ultra 5 125U × 14 processor

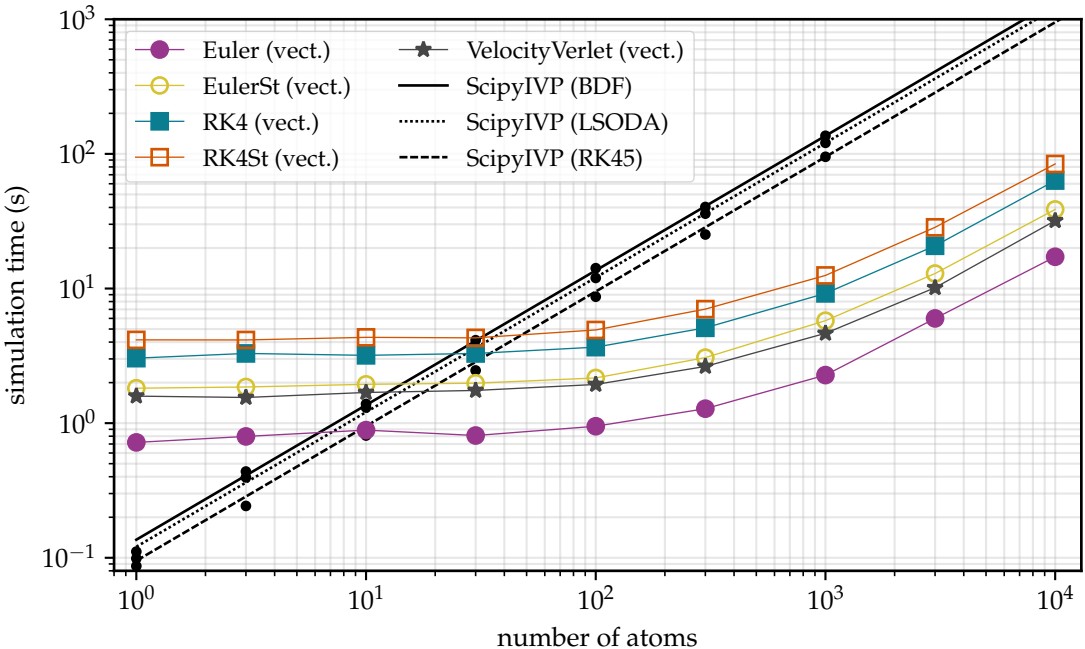

Figure 5: Computation time as a function of the number of independent simulations for different integration models in `atomsmltr`. The horizontal axis indicates the number of atoms, corresponding to independent simulations performed under identical experimental conditions but with varying initial positions and velocities. The in-house integrators (`Euler`, `EulerSt`, `RK4`, `RK4St` and `VelocityVerlet`) exploit vectorization to run simulations in parallel, whereas the scipy-based models (`ScipyIVP`) execute them sequentially. For small atom numbers, the scipy-based models are faster; however, as the number of atoms increases, the benefits of vectorization become apparent, and the `atomsmltr` models achieve superior performance.

each initialized with randomly assigned positions and velocities. For the built-in simulators, we exploit parallelization through array operations by providing the initial conditions as a vector of shape $(N, 6)$, where $N$ is the number of atoms (cf. section 3.7). In contrast, the scipy wrapper does not support such vectorization and requires an explicit iteration over each set of initial conditions. As a result, running a simulation for $N$ atoms is equivalent to executing $N$ independent simulations, all sharing the same configuration but differing in the atom's initial position and velocity.

The results of the performance comparison are shown in Figure 5. For a single simulation, the scipy-based solver is at least one order of magnitude faster than the built-in `atomsmltr` integrators. However, as the number of atoms increases, the computation time for the scipy-based solver increases linearly, as expected, since it does not exploit vectorization or parallelization. In contrast, the execution time of the `atomsmltr` simulators remains nearly constant for ensembles of up to a few thousand atoms, outperforming the scipy-based solver once more than a few tens of atoms are simulated. This behavior clearly demonstrates the advantage of the vectorization strategy implemented in `atomsmltr`. Within the `atomsmltr` integrator collection, the fastest methods are those requiring fewer operations per integration steps, such as `Euler` and `VelocityVerlet`, whereas stochastic integrators are consistently

with 16 GB or RAM.

slower than their deterministic counterparts. Finally, it should be noted that the computational speedup achieved through vectorization comes at the cost of increased memory usage, due to the manipulation of large multi-dimensional arrays.

## 5   Application example

After benchmarking `atomsmltr` on canonical examples, we now turn to more complex scenarios, where its advantages become particularly evident. We consider two cases: (i) the slowing and capture of strontium atoms in an optimized cold-atom source [14], and (ii) the launch of an atomic cloud in an atomic fountain in the so-called (1,1,1) configuration [16].

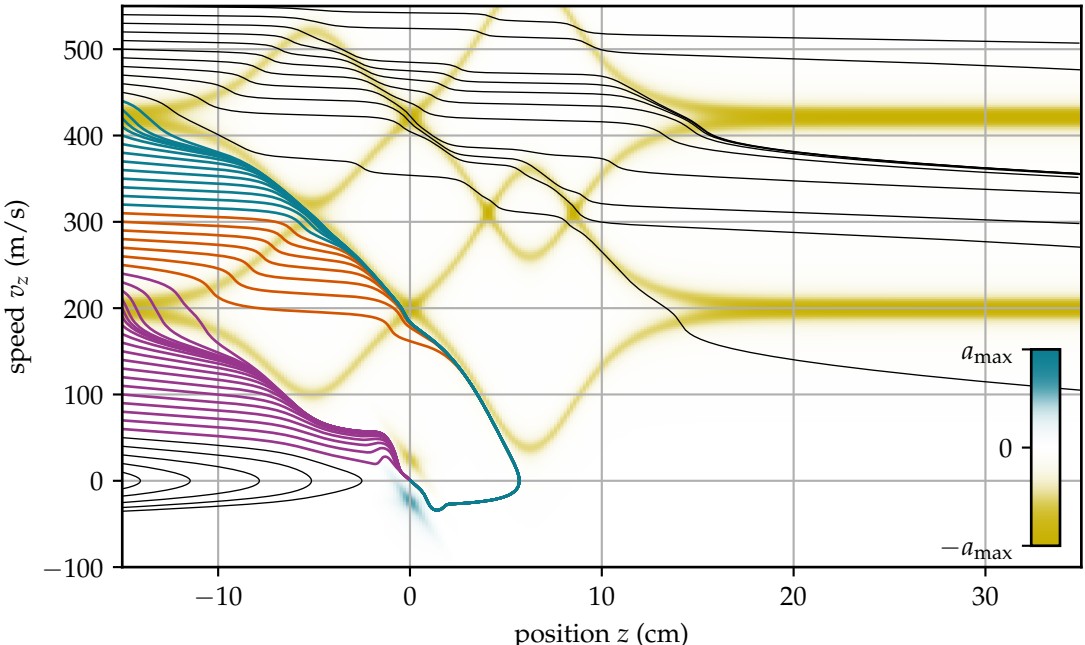

Figure 6: Simulated phase-space trajectories of strontium atoms propagating from an oven located at $z = -15$ cm, reproducing the configuration described in Figure 5 of [14], using `atomsmltr`. The atoms are slowed by a compact Zeeman slower using a two-frequency laser beam along the $-z$ direction and subsequently trapped in a 2D-MOT aligned with the $y$ axis. The slowing beam contains two frequency components, red-detuned from the 461 nm cooling transition by $\delta_1 = -14\,\Gamma$ and $\delta_2 = -30\,\Gamma$, respectively. Purple trajectories correspond to atoms decelerated by the $\sigma-$ component of the slowing beam between the oven and the first stacks of magnets, which are then trapped in the 2D-MOT. Orange trajectories represent atoms decelerated by the $\sigma+$ component in the region between the four stacks of magnets; these atoms pass once through the 2D-MOT region with positive velocity without being captured, then fall back and are eventually trapped. Blue trajectories indicate atoms initially decelerated by the $\sigma-$ component of the second frequency in the Zeeman beam between the oven and the first stacks of magnets, which then merge with the orange trajectories. The acceleration experienced by the atoms in this configuration of magnetic fields and laser beams, also computed using `atomsmltr`, is shown in the background.

## 5.1   Illustration #1: high-flux strontium source

In Ref. [14], one of us realized a cold strontium atomic source efficiently cooling and trapping a large fraction of the thermal atomic beam emitted by an oven. This was achieved using a compact Zeeman slower followed by a 2D-MOT, both relying on magnetic fields generated by permanent magnets. The configuration introduced several novel features. First, the Zeeman slower decelerates atoms emitted from the oven using both polarization components of the slowing light. Second, atoms are captured by the 2D-MOT not only directly when propagating from the oven, but also from the opposite direction, after having passed once through the MOT region. Third, an optimized magnetic field profile increases the 2D-MOT atomic flux by mitigating losses due to the transverse expansion of the thermal beam. Finally, a two-frequency configuration of the slowing beam extends the capture velocity of the Zeeman slower.

   All these features can be observed in Figure 6, which reproduces with excellent agreement the results reported in Figure 5 of [14]. In this configuration, we consider strontium atoms cooled on the $^1S_0 \rightarrow {}^1P_1$ transition at 461 nm. The atoms propagates one-dimensionally from an oven located at $z = -15$ cm and interact with a Zeeman slowing beam, which can operate either in a single-frequency mode (detuning $\delta_1 = -14\,\Gamma$, saturation parameter $s_1 = 1.5$) or in a two-frequency mode (adding second frequency component with detuning $\delta_2 = -30\,\Gamma$, saturation parameter $s_2 = 2.1$). At z=0 two counter-propagating beams aligned along the $x \pm y$ directions form the 2D-MOT. Each beam has a $1/e^2$ waist of 12 mm and a total optical power of 80 mW. The zero of the 2D-MOT is aligned with the $y$ axis. The magnetic field for both the 2D-MOT and the Zeeman slower is generated by four stack of permanet magnets, which are modeled in the numerical simulation using the `magpylib` module. This example showcases `atomsmltr`'s capability to handle complex, realistic magnetic field configurations, as well as its powerful integration of `magpylib` objects.

## 5.2   Illustration #2: launch in an atomic fountain

The so-called (1,1,1) configuration [17], commonly used in atomic fountains, is obtained from the standard 3D MOT geometry, with beam aligned along the Cartesian axes $x$, $y$, and $z$, by a global rotation that brings one of the MOT diagonals along the laboratory vertical axis $z$. This transformation can be achieved by two successive rotations: first, a rotation around the z-axis by an angle $\phi = -\pi/4$, followed by a rotation around the y-axis by $\theta = -\arccos(1/\sqrt{3})$. The resulting overall rotation matrix $R$ is given by:

$$R = \begin{pmatrix} 1/\sqrt{6} & -1/\sqrt{2} & 1/\sqrt{3} \\ 1/\sqrt{6} & 1/\sqrt{2} & 1/\sqrt{3} \\ -\sqrt{2/3} & 0 & 1/\sqrt{3} \end{pmatrix}. \tag{7}$$

This transformation aligns the original (1,1,1) diagonal with the vertical z-axis while preserving the orthogonality of the three MOT beam axes. In this configuration the three upward beams are directed along the unit wave vectors $\vec{u}_i = \left( \sqrt{2/3}\cos\phi_i, \sqrt{2/3}\sin\phi_i, 1/\sqrt{3} \right)$ with $\phi_i = 0, 2\pi/3, 4\pi/3$, while the three corresponding downward beams propagate along $-\vec{u}_i$. By symmetry, their horizontal components cancel, so that a relative detuning between the upwards and downward sets produces a moving optical molasses purely along the vertical direction, with a corresponding net momentum transfer in the vertical direction. This geometry also leaves the vertical axis unobstructed for atomic launches, due to the absence of a stationary vertical beam pair and the consequent reduction of light-shift effects during the ballistic flight.

   Once a sufficient number of atoms has been loaded into the MOT, the magnetic field is switched off, and the light parameters are adjusted to launch the atoms upwards. The launch

mechanism relies on the moving optical molasses described above: in the (1,1,1) configuration, applying a relative detuning $\epsilon$ is applied between the three upward- and three downward-propagating beams of wave vector **k** both cools the atomic ensemble and accelerates its center of mass. The resulting maximal vertical velocity $v_z^{\max}(\epsilon)$ is then given by:

$$v_z^{\max}(\epsilon) = \frac{\epsilon}{k\cos\theta},\qquad(8)$$

which corresponds to approximately $1.35\,\mathrm{m\,s^{-1}}$ per MHz of detuning $\epsilon$ for the rubidium D2 line at $780\,\mathrm{nm}$. By adjusting the detuning, one can therefore control the launch velocity and the maximum height reached by the atomic cloud.

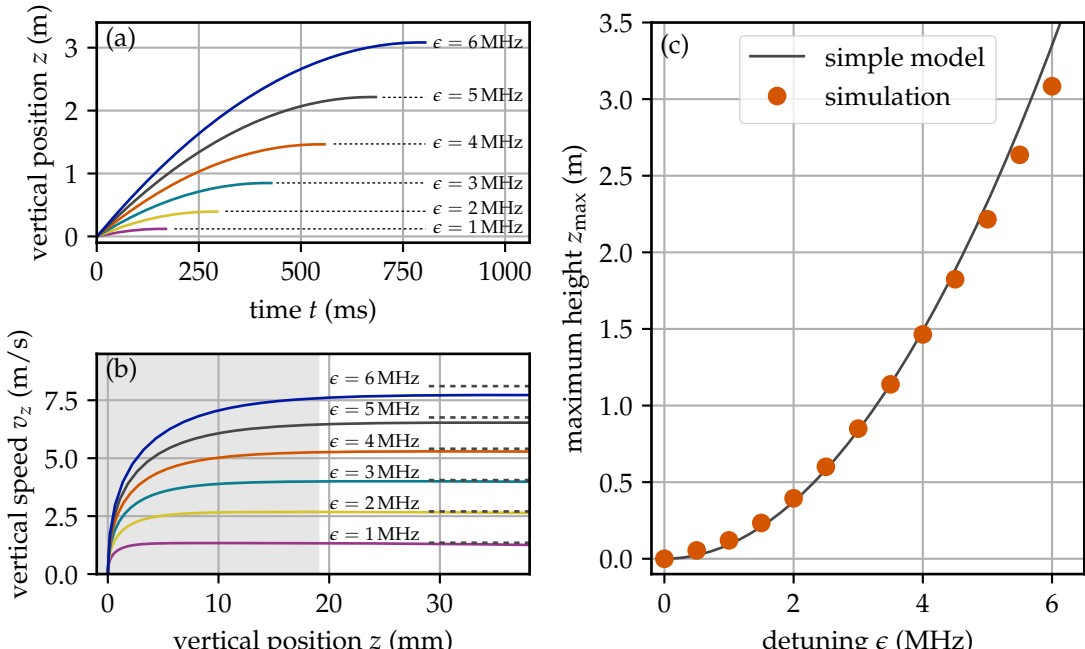

Figure 7: Launch simulation in an atomic fountain operated in the (1,1,1) configuration using `atomsmltr`. **(a) Vertical trajectories:** the vertical position as a function of time for different values of the differential frequency detuning $\epsilon$ between the upward- and downward-propagating MOT beams. Each simulation is terminated when the atom reaches its apex. **(b) Phase-space plot:** vertical velocity versus position at the beginning of the launch, *i.e.*, while the atom is still being accelerated by the moving molasses. The approximate spatial extent of the molasses is indicated by a grey area, and the theoretical maximal velocity for each detuning $\epsilon$ is shown as a black dotted line. **(c) Maximum height:** the maximum height $z_{\max}$ obtained from simulation is plotted as a function of the differential frequency detuning $\epsilon$ (circles), together with the theoretical height derived from the moving molasses velocity (black line). For large detunings, a deviation appears because the atom does not remain in the moving molasses long enough to reach the moving molasses velocity.

In Figure 7 we present the results of a simulation of atomic launch in the (1,1,1) configuration for rubidium atoms, performed on the D2 transition ($\lambda = 780\,\mathrm{nm}$). To reach the theoretical limit and accurately characterize the configuration, stochastic effects are disabled, as they can occasionally push the atoms out of the trapping region and hinder an efficient launch. The Gaussian beams have a power $P = 16.7\,\mathrm{mW}$, a $1/e^2$ waist of $22\,\mathrm{mm}$, and a detuning of $\delta = 3\,\Gamma$, arranged in the same geometry as in [16]. Atoms are initialized at rest and at

the center of the optical molasses. We simulate the launch for different values of the differential detuning $\epsilon$ between the upward and downward beams, ranging from 0.5 MHz to 6 MHz. The agreement between the theoretical predictions and the simulations for both the maximum height and the initial launch velocity is excellent, as shown on Figure 7(c). Small deviations appear at large detunings, which we attribute to the finite time the atoms spend in the moving molasses, preventing full acceleration to the maximal velocity, see Figure 7(b). This example further demonstrates the ability of `atomsmltr` to model a complex experimental setup, and quickly reveal subtle physical effects that would be neglected in a simplified description.

## 6   Conclusion

We have presented `atomsmltr`, a Python package for simulating laser cooling of neutral atoms in complex configurations of magnetic fields and laser beams. Thanks to its modular design and built-in vectorized integrators, `atomsmltr` is well suited for a wide range of simulations under realistic experimental conditions, including the optimization of atomic sources and magneto-optical trap configurations.

We have detailed the physical assumptions and models underlying `atomsmltr`, described its main components, and provided a set of examples illustrating its capabilities and performance on textbook laser cooling scenarios. This benchmarking validates the accuracy of `atomsmltr` and demonstrates its suitability for tackling more complex situations. As illustrative applications, we have simulated the slowing and capture of atoms in a realistic strontium atom source, as well as the launch of atoms in an atomic fountain. These results indicate that `atomsmltr`, in its current state of development, constitutes a valuable addition to the atomic physics simulation toolbox.

We have also discussed the current limitations of `atomsmltr`. Atomic transitions are modeled as $J = 0 \to 1$ systems, *i.e.*, with a non-degenerate ground state, and atom-light interactions are presently restricted to radiation pressure and photon scattering. While this is generally sufficient for tasks such as optimizing Zeeman slowers or simulating species such as bosonic strontium or ytterbium, more sophisticated studies will require extensions to include hyperfine structure, off-resonant light shifts, and multi-level atomic transitions.

The package remains under active development, and several new features are planned to address these limitations. Future work might include implementing optical Bloch equation solvers, extending the atomic structure beyond the $J = 0 \to 1$ approximation, and incorporating off-resonant light shifts to enable simulations of dipole trap physics. Additional laser beam geometries and magnetic field profiles can also be added, and improvements to configuration handling, such as the ability to declare and save experimental setups in a compact, human-readable format, are being considered. The modular architecture of `atomsmltr` will facilitate these developments, and we encourage users to contribute by suggesting new features[8] or by participating directly in their implementation.

## Acknowledgements

**Author contributions**   A.D. designed the core codebase of `atomsmltr`, drawing inspiration from earlier simulations performed by A.B.; A.B. and M.W. tested and benchmarked `atomsmltr`, and implemented the configurations used in this work, which are now included in the package as built-in examples; all authors contributed to the analysis of the simulation

---

[8]See https://github.com/adareau/atomSmltr/issues

results and to the writing of the manuscript.

**Funding information**  This work received funding from the French government, managed by the National Research Agency under France 2030 investment plan with the reference ANR-23-ATOM-0001. It was partly supported by the European Union's Horizon 2020 research and innovation programme (FET-Open project CRYST$^3$ N. 964531, project Qu-Test N. 101113901), Euramet (Project 23FUN02 CoCoRICO), Conseil Régional de Nouvelle Aquitaine and Naquidis Center (QPLEX project), the EUR Light S&T Graduate Program (PIA3 Program "Investment for the Future", ANR-17-EURE-0027), and the Idex of the University of Bordeaux (Research Program "GPR Light"). Qu-Test Project has received funding from the European Union's Horizon Europe—The EU research and innovation program under the Grant Agreement 101113901.

# A   Getting started

## A.1   Installation

### A.1.1   Install latest stable release (recommended)

The `atomsmltr` package is publicly available on the Python Package Index[9] and can be installed directly using `pip`:

```
pip install atomsmltr
```

To make use of the optional `magpylib` integration, the corresponding package must also be installed:

```
pip install magpylib
```

### A.1.2   Install versions under development (advanced user)

The source code of `atomsmltr` is available on GitHub[10]. Advanced users interested in contributing to the development of `atomsmltr`, or wishing to access the latest updates, can install the package directly from the GitHub repository. To do so, clone the repository and switch to the desired branch (here, `testing`):

```
git clone https://github.com/adareau/atomSmltr.git
cd atomSmltr
git checkout testing
```

We strongly encourage using a virtual environment, which can be created as follows:

```
python3 -m venv __venv__
source ./__venv__/bin/activate
```

Finally, install the package locally from the `atomsmltr` directory:

```
pip install .
```

In our Git development workflow, three main branches are maintained: `main` for stable releases; `testing` for development versions expected to work reliably under most conditions; `devel` for implementating new and more experimental features. Users who wish to access the latest functionalities are encouraged to use the `testing` branch.

## A.2   Read the manual

Comprehensive online documentation for `atomsmltr` is available, including a collection of use cases, hands-on examples, and detailed reference material. It can be accessed at:

https://atomsmltr.readthedocs.io

The example notebooks presented in the documentation are also available directly in the GitHub repository, in the form of Jupyter notebooks:

atomsmtlr/docs/_notebook_examples

---

[9]https://pypi.org/project/atomsmltr/
[10]https://github.com/adareau/atomSmltr

### A.3  Running a first experiment

A short code snippet illustrating the use of `atomsmltr` is shown below; it simulates the dynamics of a ytterbium atom in a one-dimensional optical molasses operating on the main transition at 399 nm.

```python
import numpy as np
import matplotlib.pyplot as plt

from atomsmltr.environment import PlaneWaveLaserBeam
from atomsmltr.atoms import Ytterbium
from atomsmltr.simulation import Configuration, RK4

# - setup atom
atom = Ytterbium()
main = atom.trans["main"]  # get transition, to help setting up lasers

# - setup laser
laser_1 = PlaneWaveLaserBeam()
laser_1.direction = (0, 0, 1)
laser_1.set_power_from_I(main.Isat)  # set power to reach Isat
laser_1.tag = "las1"

laser_2 = laser_1.copy()  # create a copy
laser_2.direction = (0, 0, -1)  # propagating in opposite direction
laser_2.tag = "las2"

# - config
config = Configuration()
config.atom = atom
config += laser_1, laser_2
config.add_atomlight_coupling("las1", "main", -0.5 * main.Gamma)
config.add_atomlight_coupling("las2", "main", -0.5 * main.Gamma)

# - simulation
sim = RK4(config=config)
t = np.linspace(0, 0.1, 3000)  # timesteps for integration
u0 = (0, 0, 0, 0, 0, 100)  # atom starts with vz=100m/s
res = sim.integrate(u0, t)

# plot
fix, axes = plt.subplots(1, 2, figsize=(8, 3), tight_layout=True)
axes[0].plot(res.t * 1e3, res.y[2])
axes[0].set_ylabel("z␣(m)")
axes[1].plot(res.t * 1e3, res.y[5])
axes[1].set_ylabel("vz␣(m/s)")
for ax in axes:
    ax.set_xlabel("t␣(ms)")
    ax.grid()
plt.show()
```

### A.4 Reproducing examples from this paper

`atomsmltr` is distributed with a collection of configuration files reproducing all the examples presented in this article. These files serve as a convenient starting points for users to explore and experiment with simulations. Below is a list of the available configurations and their corresponding figures from the present work:

**Figure 2: one-dimensional MOT for ytterbium atoms**
`from atomsmltr.examples.atomsmltr2025 import config_1D_MOT_Yb`
➥ ytterbium, 2 × plane wave laser beams at 399 nm with $s_0 = 0.02$ and $\delta = -\Gamma/2$, counter-propagating along $z$ with ($R$) polarization, magnetic field gradient along $z$ with various slopes.

**Figure 3: three-dimensional optical molasses for ytterbium atoms**
`from atomsmltr.examples.atomsmltr2025 import config_Doppler_limit`
➥ ytterbium, 6 × plane wave laser beams at 399 nm with $s_0 = 0.05$ and various $\delta$, counter-propagating along $x$, $y$, $z$.

**Figure 4(a): one-dimensional optical molasses for rubidium atoms**
`from atomsmltr.examples.chen2021 import config_1D_molasses`
➥ rubidium, 2 × Gaussian laser beams at 780 nm with $w = 1.41$ cm, $P = 10$ mW and $\delta = -2\pi \times 6$ MHz, counter-propagating along $z$.

**Figure 4(b): one-dimensional MOT for strontium atoms**
`from atomsmltr.examples.chen2021 import config_1D_MOT`
➥ strontium, 2 × Gaussian laser beams at 461 nm with $w = 1.41$ cm, $P = 30$ mW and $\delta = -2\pi \times 12$ MHz, counter-propagating along $z$ with ($L$) polarization, magnetic quadrupole with strong axis along $z$ and a slope of $0.15\,\text{T}\,\text{m}^{-1}$.

**Figure 5: three-dimensional MOT for ytterbium atoms**
`from atomsmltr.examples.atomsmltr2025 import config_3D_MOT_Yb`
➥ ytterbium, 6 × Gaussian laser beams at 399 nm with $w = 22$ mm, $P = 17$ mW and $\delta = -\Gamma$, counter-propagating along $x$, $y$, $z$ with ($R$) polarization along $x$ and $y$ and ($L$) polarization along $z$, magnetic quadrupole with strong axis along $z$ and a slope of $30\,\text{G}\,\text{cm}^{-1}$.

**Figure 6: high-flux atomic strontium source**
`from atomsmltr.examples.feng2024 import config_asymmetric_field_2`
➥ strontium, configuration for an atomic source with a Zeeman slower and 2D-MOT, with two frequencies in the Zeeman slower and an asymmetric MOT field, see [14] for more details.

**Figure 7: atomic fountain for rubidium**
`from atomsmltr.examples.atomic_fountain import config`
➥ rubidium, configuration for an atomic fountain in the (1,1,1) configuration, see main text and [16] for more details.

# B   Comparison with other packages

We provide here a non-exhaustive list of software packages dedicated to atomic physics simulations and calculations, along with a brief comparison to `atomsmltr`. We encourage readers to explore these alternative tools, which offer features that complement those of `atomsmltr`.

- `QuTiP` [7,8] (https://qutip.org/): QuTiP is a highly versatile and user-friendly Python toolbox for simulating complex quantum systems, whereas `atomsmltr` is currently limited to semi-classical approximations. However, QuTiP is not optimized for simulating the motion of particles in complex configurations of magnetic fields and laser beams, giving it a scope distinct from that of `atomsmltr`.

- `atomECS` [9] (https://github.com/TeamAtomECS/AtomECS): atomECS shares the same objectives as `atomsmltr`, which is why we benchmarked our simulations against it in the present work. Implemented in Rust, atomECS may offer a performance advantage over `atomsmltr`, albeit at the expense of user-friendliness. Unlike `atomsmltr`, atomECS was designed to simulate multiple atoms evolving in the same configuration, thereby enabling the study of collective effects. In terms of underlying physical models, atomECS and `atomsmltr` are quite similar. However, `atomsmltr` offers advantages in ease of use and in handling complex configurations of magnetic fields and laser beams, thanks to its Python implementation, integration with magpylib, advanced polarization handling, and related features.

- `ARC` [10] (https://arc-alkali-rydberg-calculator.readthedocs.io): ARC is a Python package for calculating single- and two-atoms properties, with a primary focus on Rydberg physics. We include it here as part of the broader atomic physics Python ecosystem, although its scope is clearly distinct from that of `atomsmltr`.

- `PyLCP` [11] (https://python-laser-cooling-physics.readthedocs.io): among the Python libraries we have considered, PyLCP is perhaps the one whose scope most directly overlaps with that of `atomsmltr`. Although we did not test it extensively, PyLCP appears to allow more detailed modeling of light-matter interaction effects through the definition of Hamiltonians and their solution via optical Bloch equations. In contrast, `atomsmltr`'s modular approach to defining experimental configurations is better suited for complex simulations, particularly for species such as bosonic ytterbium or strontium, where effects beyond radiation pressure and photon scattering (*e.g.*, optical pumping) play a minor role. Nevertheless, it would be interesting in future developments of `atomsmltr` to explore whether the simulation framework developed for PyLCP could be integrated with `atomsmltr`'s configuration management to extend the capacities of the ecosystem.

## C  Laser conventions

### C.1  Laser propagation convention

Within the `atomsmltr` package, laser beams are represented by a dedicated `LaserBeam` class. When initializing a `LaserBeam` object, the propagation direction must be specified, which can be done in two ways, see Figure 8(a):

1. As a **vector** $\vec{u}$, defining the propagation axis and direction of the laser.

2. As a **tuple** $(\theta, \phi)$, where $\theta$ and $\phi$ denote the **polar** and **azimuthal** angles, respectively, defining the orientation of the laser's propagation axis.

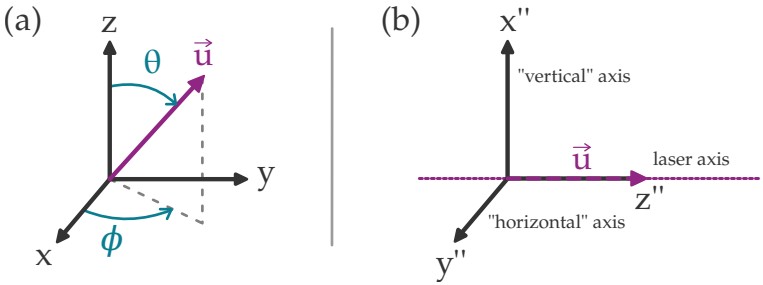

Figure 8: Convention for laser propagation. (a) The laser propagation is defined by a direction vector $\vec{u}$, which can be parametrized by two angles, $\theta$ and $\phi$, corresponding to the polar and azimuthal angles of $\vec{u}$ in the laboratory frame. (b) In the *laser frame*, the laser propagates along the $z''$ axis, while the $x''$ and $y''$ axes are referred to as the "vertical" and "horizontal" directions, respectively.

Some laser properties, such as polarization, are defined in the *laser frame*. In the following, we denote $\mathcal{R}$ the laboratory frame and $\mathcal{R}''$ the laser frame (see Figure 8(b)). The laser frame is defined such that the laser propagation unit vector coincides with the $z''$ axis, *i.e.*, $\vec{e}_z'' \equiv \vec{u}$. We denote by $(x'', y'', z'')|_{\mathcal{R}''}$ the coordinates in the laser frame. Since this condition alone does not uniquely determine $\mathcal{R}''$, the transformation from the laboratory frame $\mathcal{R}$ to the laser frame $\mathcal{R}''$ is defined in two steps, as detailed below:

1. **First step:** $\mathcal{R} \to \mathcal{R}'$

   We first perform a rotation about $\vec{e}_z$ by an angle $\phi$. The coordinates $(x', y', z')|_{\mathcal{R}'}$ in the new frame are then given by

   $$\begin{cases} x' = & x\cos(\phi) & + & y\sin(\phi) \\ y' = & -x\sin(\phi) & + & y\cos(\phi) \\ z' = & z \end{cases} \tag{C.1}$$

2. **Second step:** $\mathcal{R}' \to \mathcal{R}''$

   Next, we perform a rotation about $\vec{e}_y'$ by an angle $\theta$. After this second rotation, $\vec{e}_z''$ coincides with the propagation direction $\vec{u}$. The coordinates $(x'', y'', z'')|_{\mathcal{R}''}$ in the new frame are then given by

   $$\begin{cases} x'' = & x'\cos(\theta) & - & z'\sin(\theta) \\ y'' = & y' \\ z'' = & x'\sin(\theta) & + & z'\cos(\theta) \end{cases} \tag{C.2}$$

Consequently, the coordinates in the laser frame, $(x'', y'', z'')|_{\mathcal{R}''}$ are related to those in the laboratory frame, $(x, y, z)|_{\mathcal{R}}$, as follows:

$$
\begin{cases}
x'' = & x\cos(\theta)\cos(\phi) & + & y\cos(\theta)\sin(\phi) & - & z\sin(\theta) \\
y'' = & -\ x\sin(\phi) & + & y\cos(\phi) & & \\
z'' = & x\sin(\theta)\cos(\phi) & + & y\sin(\theta)\sin(\phi) & + & z\cos(\theta)
\end{cases}
\tag{C.3}
$$

or equivalently, the inverse transformation reads

$$
\begin{cases}
x = & x''\cos(\theta)\cos(\phi) & - & y''\sin(\phi) & + & z''\sin(\theta)\cos(\phi) \\
y = & x''\cos(\theta)\sin(\phi) & + & y''\cos(\phi) & + & z''\sin(\theta)\sin(\phi) \\
z = & -\ x''\sin(\theta) & & & + & z''\cos(\theta)
\end{cases}
\tag{C.4}
$$

### C.2 Laser polarization naming

As explained in the previous section, the polarization is defined in the laser frame. For linear polarizations, we define the vertical direction along the $x''$ axis, and the horizontal direction along the $y''$ axis. For circular polarizations, we adopt the source-point-of-view convention, as illustrated in Figure 9. In the following, we denote the polarization states $(H)$, $(V)$, $(R)$ and $(L)$ as horizontal, vertical, right-circular, and left-circular, respectively.

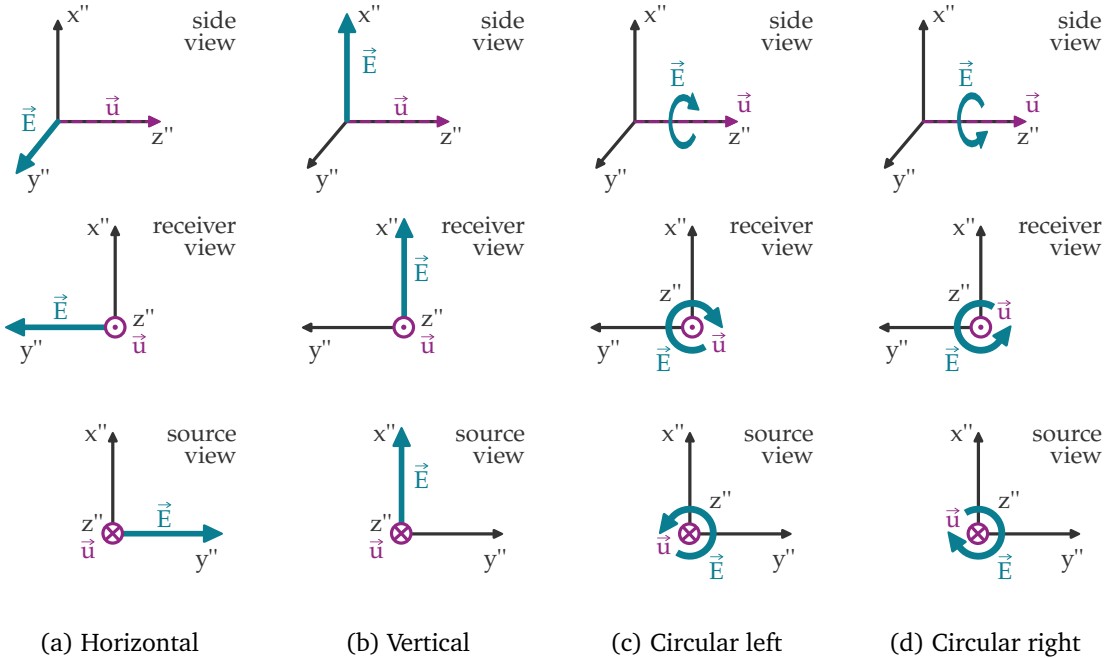

(a) Horizontal      (b) Vertical      (c) Circular left      (d) Circular right

Figure 9: Naming convention for laser polarization in `atomsmltr`, using the source-point-of-view convention.

### C.3 Quantization axis convention

In `atomsmltr`, the quantization axis is always defined as aligned with the local magnetic field $\vec{B}$. This allows projecting the laser polarization states $(H, V, R, L)$, which are defined independently of the magnetic field, onto the $\pi$, $\sigma+$ and $\sigma-$ components. In the next section, we present a formalism for calculating this decomposition for arbitrary orientations of the magnetic field and laser polarization. Here, we illustrate a few examples for circular polarizations in the special case where $\vec{u}$ and $\vec{B}$ are collinear. When $\vec{u}$ and $\vec{B}$ are co-propagating

(Figure 10a,10b), left- and right-circular polarizations correspond to $\sigma-$ and $\sigma+$ couplings, respectively. When $\vec{u}$ and $\vec{B}$ are counter-propagating (Figure 10c,10d), left- and right-circular polarizations correspond to $\sigma+$ and $\sigma-$ couplings, respectively. Finally, a $\pi$ coupling corresponds to a linear polarization aligned with the magnetic field axis.

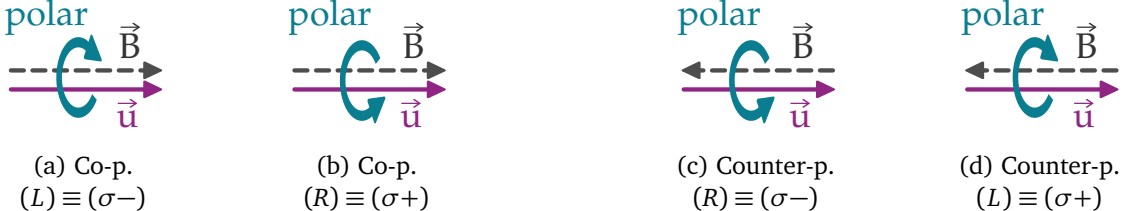

(a) Co-p.
$(L) \equiv (\sigma-)$

(b) Co-p.
$(R) \equiv (\sigma+)$

(c) Counter-p.
$(R) \equiv (\sigma-)$

(d) Counter-p.
$(L) \equiv (\sigma+)$

Figure 10: Correspondence between laser polarization and the atomic quantization axis for laser beams propagating along the magnetic field, either co-propagating (a,b) or counter-propagating (c,d).

## C.4  Polarization vector formalism

To compute the projection of an arbitrary laser polarization state onto the atomic quantization axis, we adopt a general vector formalism inspired by the Bloch sphere. The polarization is represented by a vector $\vec{p}$, specified by its polar and azimuthal angles, $u$ and $v$, in the *laser frame* (see previous section). This sphere-based representation is illustrated in Figure 11.

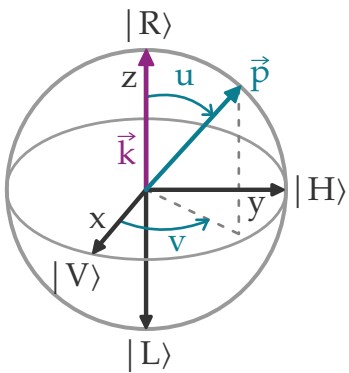

Figure 11: Representation of the polarization vector on a sphere. The laser wavevector is denoted $\vec{k}$ ($\vec{k} = k\vec{u}$). In the laser frame, the polarization vector $\vec{p}$ is defined by its polar and azimuthal angles, $u$ and $v$. On the sphere, circular polarizations $(R, L)$ lie at the poles, while linear polarizations lie on the equator.

The polarization vector is defined as follows:

- **Right-circular polarization** $|R\rangle$: the polarization vector points along the laser propagation direction, *i.e.*, $\vec{p}$ is aligned with $\vec{k}$, corresponding to the north pole of the sphere.

- **Left-circular polarization** $|L\rangle$: the polarization vector points opposite to the laser propagation direction, corresponding to the south pole of the sphere.

- **Linear polarizations**: the polarization vector $\vec{p}$ lies on the equator of the sphere.

- **Horizontal polarization** $|H\rangle$: the polarization vector aligns with the $y$ axis on the equator.

- **Vertical polarization** $|V\rangle$: the polarization vector aligns with the $x$ axis on the equator.

For self-consistency, the formalism satisfies the following relations for the polarization state $|p\rangle$ :

$$|p\rangle = e^{-iv}\cos(u/2)|R\rangle + e^{iv}\sin(u/2)|L\rangle \tag{C.5}$$

$$|V\rangle = \frac{1}{\sqrt{2}}(|R\rangle + |L\rangle) \tag{C.6}$$

$$|H\rangle = \frac{i}{\sqrt{2}}(|L\rangle - |R\rangle) \tag{C.7}$$

## C.5 Deriving projections in arbitrary geometries

Using this formalism, the polarization state $|p\rangle$ can be decomposed into its $\pi$, $\sigma+$, and $\sigma-$ components relative to a given magnetic field $|B\rangle$, which defines the quantization axis. In the following, we work in the *laser frame*, where the $z$-axis is aligned with the laser wavevector $\vec{k}$.

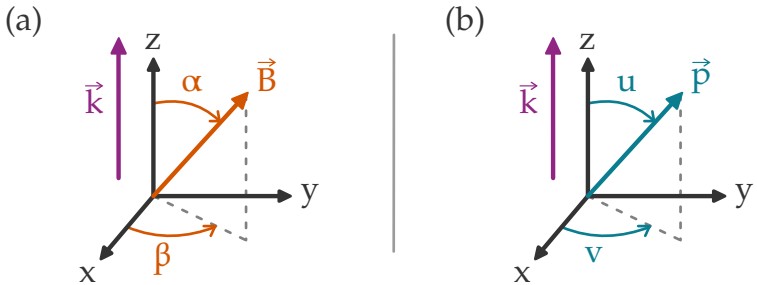

Figure 12: Angle conventions for the magnetic field (a) and polarization vector (b), both defined in the *laser frame*.

The orientation of the polarization $|p\rangle$ and the magnetic field vector $|B\rangle$ are specified by the angles $(u, v)$ and $(\alpha, \beta)$, respectively, as illustrated in Figure 12. The problem be formulated as follows:

- in the *laser basis* $(x, y, z)$, the polarization state can be expressed as

  $$|p\rangle = e^{-iv}\cos(u/2)|R\rangle + e^{iv}\sin(u/2)|L\rangle,$$

  which can be further decomposed onto $|x\rangle = |V\rangle$ and $|y\rangle = |H\rangle$ using $|R\rangle = (|x\rangle + i|y\rangle)/\sqrt{2}$ and $|L\rangle = (|x\rangle - i|y\rangle)/\sqrt{2}$.

- in the *magnetic field* basis $(x', y', z')$, where $z'$ is aligned with $\vec{B}$, the polarization states are defined as $|\pi\rangle = |z'\rangle$ and $|\sigma\pm\rangle = (|x'\rangle \pm i|y'\rangle)/\sqrt{2}$.

Using the formulas above, it is possible to (i) decompose the polarization state $|p\rangle$ onto the $|x\rangle$ and $|y\rangle$ states, and (ii) compute its projection onto the $|pi\rangle$ and $|\sigma\pm\rangle$ states. This procedure yields the following expressions:

$$|p\rangle = \left\{ e^{-iv}\cos(u/2) + e^{iv}\sin(u/2) \right\} /\sqrt{2}\, |x\rangle$$
$$+ i\left\{ e^{-iv}\cos(u/2) - e^{iv}\sin(u/2) \right\} /\sqrt{2}\, |y\rangle$$

$$|x\rangle = (\cos\beta\cos\alpha + i\sin\beta)/\sqrt{2}\, |\sigma+\rangle$$
$$+ (\cos\beta\cos\alpha - i\sin\beta)/\sqrt{2}\, |\sigma-\rangle$$
$$+ \cos\beta\sin\alpha\, |\pi\rangle$$

$$|y\rangle = (\sin\beta\cos\alpha - i\cos\beta)/\sqrt{2}\,|\sigma+\rangle$$
$$+ (\sin\beta\cos\alpha + i\cos\beta)/\sqrt{2}\,|\sigma-\rangle$$
$$+ \sin\beta\sin\alpha\,|\pi\rangle$$

Using these formulas, the projections of the polarization state onto the magnetic field basis can be computed as $\langle p|\pi\rangle$, $\langle p|\sigma+\rangle$, and $\langle p|\sigma-\rangle$.

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
