# Peer review of "atomSmltr: a modular Python package to simulate laser cooling setups"

_SciPost Physics Codebases_

## Round 1 · Referee Report · Elliot Bentine (Referee 1) · 2026-1-7

Strengths

  1. The submission is well formatted and clearly written; this applies to both the paper and the documentation. The documentation is clearly presented, well formatted and easy to navigate.

  2. The physical systems simulated are workhorse techniques that form the starting point for many experiments in the field, and thus the atomic and laser physics community will benefit from a high-quality package such as atomSmltr to simulate these systems. The choice of language and clarity of examples will give the package broad appeal and uptake.

  3. Section 2.2 of the userguide clearly states the intended context and limitations of the package. A very good job has been done here, and this will greatly aid others in the field adopting atomSmltr for their own work, as they should be able to understand the impact of the program's choice of model on the generated results.

  4. The authors have taken great care to clearly define all conventions used (Appendix C), with clear diagrams and concise wording. This will be a valuable resource for anyone seeking to extend or modify the package.

  5. The application examples are worked through in great detail and provide an excellent reference point for others seeking to use the package. Existing results in the literature are reproduced which gives confidence in the accuracy of the package.

  6. I greatly appreciated the explicit and detailed instructions in A.4 for reproducing the examples from the paper which will be useful to others seeking to reproduce the results.

  7. The program is well tested through unit tests (pytest), which all succeed at time of writing (72 tests).

Weaknesses

  1. The package is limited to a simple two-level model of the underlying atom-laser interaction. Nonetheless, the authors are clear on this limitation and those in the field will understand the impact of this on generated results.

  2. atomSmltr cannot consider atom-atom effects, such as long-ranged photon rescattering interactions. These effects can become significant in 3D magneto-optical traps (where photon rescattering limits the achievable phase space density), but the computational complexity of simulating these long-ranged forces means that very few codes have considered them in detail ( e.g. https://arxiv.org/abs/2003.10919 ). As before, the authors have clearly stated this limitation, and it does not diminish the utility of atomSmltr for investigating a range of cold atom sources such as 2D MOTs.

  3. The package treats the interaction with each laser separately. As noted by the authors, this is valid in the weak saturation regime (s<<1). The Application Example in 5.1 is a high-flux strontium source with saturation parameters of s=1.5 and s=2.1. Due to the symmetry of the cooling beams and magnetic field, atoms travelling close to the z-axis will interact strongly with multiple beams. Treating atom-beam interactions independently will lead to an over-estimate of the total scattering force (see, for example, Appendix A of https://arxiv.org/abs/2105.06447 for a detailed treatment).

  4. It would be good to see continuous integration on the main repository; the authors have already written the tests, having a pipeline to run these and clearly output the results each time a contribution is merged will build user confidence.

  5. The comparison with other packages sometimes omits relevant information or mis-states their capabilities. I will comment in particular on the comparison to atomecs, because I am a developer of that package and thus familiar with it: "In terms of underlying physical models, atomECS and atomsmltr are quite similar. However, atomsmltr offers advantages in ease of use and in handling complex configurations of magnetic fields and laser beams, thanks to its Python implementation, integration with magpylib, advanced polarization handling, and related features." Elsewhere in the paper the authors note that AtomECS uses a multi-beam rate equation model not implemented in atomSmltr, but this statement is not present in the comparison (it would also be good to include a citation here to Hanley et al, Journ. Mod. Opt. 65, 2018, which describes a similar approach). AtomECS includes other features like dipole trapping, magnetic trapping, recoil cooling limits, and to some extent atom-atom collision frameworks for evaporative cooling, rf-dressed potentials and the effect of off resonant light shifts on laser cooling. It also allows specification of arbitrary magnetic fields through precalculated grids (although this is by no means as user friendly as the direct magpylib integration presented in atomSmltr, which is very handy!). If the authors enumerate atomSmltr's additional capabilities for fields and polarisations, an even and useful comparison should mention AtomECS' other features too.

  6. Fig 5 evaluates the benchmarks for the different integrators and compares the time taken for 1000 steps, showing RK4 as slowest and Euler as fastest. But this is an unfair comparison, because an RK4 integrator can use a larger effective timestep size and therefore requires less integration steps to simulate a desired simulation duration (compared to Euler, where the errors are poor).

  7. One of the examples in the code documentation shows how to use the package to generate a thermal beam. Theoretical expressions of velocity distributions after travelling through a collimating tube are included without reference and may be incorrect or the scenario may be unrealistic; collimated thermal beams produced by free molecular flow through a tube with specular reflections typically follow the $j(\theta)$ distribution, see e.g. Clausing J. Vac. Sci. Technol. 8. It might be worth adding a statement that this is a simplified example.

Report

It's a very nicely prepared codebase that will definitely be of use to the community. My 'weakness' text only appears longer only because there is always more detail to explain when critiquing (and the strengths are good, so there is hardly more detail I can add there).

The paper meets all the acceptance criteria: (i) benchmarking tests are provided; (ii) example applications are discussed in detail; (iii) programming standards are to a high level; (iv) the paper describes the context and limitations of the program, and (v) the value to the community; (vi) the documentation is of excellent standard and well presented. The code is also released under a suitable license (GPL-3) and is self-contained.

Installation instructions are clear and straightforward, with distribution/installation via pip. The documentation is clearly presented, well formatted and easy to navigate. Conventions and coordinate systems are clearly defined which will help users and future extension.

This paper will be an excellent addition to SciPost Physics Codebases.

Requested changes

  1. Figure 6 of the Application Example reproduces the results of Figure 5 from [14], which is a numerical simulation of the same system. There is overlap between authors of the papers which are two years apart; is there any overlap between the code used for [14] and the current paper? From the acknowledgements, it appears atomSmltr may have arisen from an adaption of this earlier work? If so, this should be made clearer or the wording adjusted (it is currently presented as validating atomSmltr on a more complex example 'with excellent agreement'). Note this is also the scenario in which the simulated model will have limitations (overestimating the 2D mot beam force, as noted in point 3 of weaknesses listed in the report).

  2. Regarding listed Weakness #6 (benchmarking graph); I was wondering what the benchmark would look like as a comparison between integrators to simulate a desired duration at a defined threshold error level - doing this drops the number of steps required for RK4 and it may well become the faster of the available integrators (the current graph shows no advantage to an RK4 scheme, which downplays an otherwise nice advantage of atomSmltr!). In real terms, using RK4 may improve the performance of the package, and it would be useful for users to see that. This is a suggestion to improve the paper, not a requested change.

  3. Regarding Weakness #5, I hope the authors consider the points I've raised regarding comparison to other packages. I have my own biases here, as an author of atomecs. I would also like to note that I messaged the corresponding author when atomSmltr appeared on arxiv, and we have had an amiable discussion about this.

  4. In Section 2.3, it would be useful to include a clear statement regarding the capabilities of atomSmltr for simulating recoil limited cooling such as in narrow-linewidth MOTs (2.3.3 mentions 'the current implements assume that a sufficiently large number of photons are scattered during each integration step' and that atomSmltr does not 'explicitly [draw] and [sum] the contribution of each scattered photon').

Recommendation

Ask for minor revision

---

## Editorial Decision

in_refereeing